# Habitat Mosaics of Sand Steppes and Forest-Steppes in the Ipoly Valley in Hungary



**Ildikó Járdi** [1,*][iD]**, Dénes Saláta** [1]**, Eszter S.-Falusi** [1]**, Ferenc Stilling** [1]**, Gergely Pápay** [1]**, Zalán Zachar** [1]**, Dominika Falvai** [1]**, Péter Csontos** [2]**, Norbert Péter** [1] **and Károly Penksza** [1]

[1]   Faculty of Agricultural and Environmental Sciences, Szent István University, Páter Károly u. 1., H-2100 Gödöllő, Hungary; Salata.Denes@szie.hu (D.S.); falueci@gmail.com (E.S.-F.); stillingf@gmail.com (F.S.); geri.papay@gmail.com (G.P.); kereskenyi1@gmail.com (Z.Z.); domi.falvai@gmail.com (D.F.); peter.norbert87@gmail.com (N.P.); penksza@gmail.com (K.P.)

[2]   Centre for Agricultural Research, Institute for Soil Science and Agricultural Chemistry, Herman Ottó út 15, H-1022 Budapest, Hungary; cspeter@mail.iif.hu

[*]   Correspondence: ildikojardi@gmail.com

**Abstract:** The present study focuses on the mosaic-like occurrences of patches of steppes and fore-steppes in the Pannonian forest-steppe zone. We present the current vegetation, which is maintained including by human landscape use, i.e., grazing and mowing. The area is complex and for this reason it shows the changes in the landscape and differences in the vegetation more diversely. We wanted to answer the questions: Do sand steppes and forest-steppes occur in the Ipoly Valley and what location? What kind of environmental effects influence the species composition on these areas? Besides classic habitat mapping, are the satellite data from Sentinel-2A useful for distinction of different areas? Comparison of vegetation patches was based on the Hungarian habitat classification system (ÁNÉR). Based on satellite images, quantile data of the Normalized Vegetation Index (NDVI) were used for comparison. Based on the result, water bodies and urban areas are clearly distinguishable from other natural habitats. In some natural vegetation types, we found visible differences, such as grasslands, i.e., sandy steppe meadows and shrubby, woody vegetation patches. Sandy vegetation mainly grows on calcareous soils, which appear to be mosaic-like in the landscape on raised alluvials on the patches of past islands and reefs. From open to continuous closed grasslands, these vegetation types mainly grow on lithosoils. New occurrences of Pannonian sandy vegetation were discovered. In the sandy areas along the Ipoly Valley, open sandy grasslands were found, which is where the northernmost known occurrences of this vegetation type are. Besides common sandy grassland species, the vegetation also contains herbs that are typical in loess-grasslands and it is maintained by grazing, similarly to the eastern Pannonian area. This type of grazing can be useful when maintaining the mosaic-like appearance and diversity of the vegetation.

**Keywords:** forest-steppes; sandy grassland; grazing-mowing; NDVI; Sentinel-2A

## 1. Introduction

The Pannonian-Pontic environmental zone (PAN) occupies the major part of the Carpathian Basin. The area is characterized by natural forest-steppe and steppe vegetation [1,2].

Numerous studies were carried out investigating the sandy areas of the Pannon region. The first significant review was published by Zólyomi [3]. Szujkóné–Lacza [4] assembled and reviewed the literature and collections of the Danube-Tisza Interfluve, while also using the data of the Botanical Department of the Hungarian Natural History Museum. However, surveys usually dealt with the central region of Hungary, which includes the most natural (i.e., relatively intact) parts. Owing to these studies, the temporal structure, the distinguishable aspects, and aspect-forming species of *Festucetum vaginatae* Simon 2000 are well known [5].

The zonal arrangement of the soil types and climate, which are characteristic of the eastern part of the continent, disbands completely and gives way to a mosaic-like landscape in the Carpathian Basin [6–8].

Both climazonal and edaphic mosaic habitats from steppe patches to forested areas [9–11], developed in sandy soil, which was altered and restricted as a result of landscape managements [12,13].

In these areas with calcareous soil in the central parts of the Carpathian Basin, the environmental factors are mosaic-like as well [14–19]. The present study examines the occurrences of the steppe and forest-steppe vegetation on the edge of the Pannonian forest-steppe region. The river Ipoly is one of the last rivers that have been preserved in their natural condition and have not been affected by water flow regulations. Not surprisingly, the Ipoly Valley is a protected nature reserve area of national significance, since it is part of the Danube-Ipoly National Park. In addition, it is also a nature conservation area (HUDI20026) and bird sanctuary (HUDI10008) and is subject to the Ramsar Convention in order to protect the migratory aquatic birds [20,21].

Despite its linear nature, Ipoly Valley harbors especially mosaic-like vegetation, mainly because it is a non-regulated, natural watercourse [22]. Close relations between the soil moisture level and degree of vegetation heterogeneity were also detected in other water-courses in the Pannonian region [23]. Based on earlier examinations of the Ipoly Valley, the changes of the ground water table clearly affect the spatial arrangement of vegetation types [24]. With the increased advance of agriculture, most of the grasslands were drained and plowed. The area along the Ipoly offers an ideal research terrain for studying the effects of various environmental factors such as soil on the distribution of native species. For these reasons, further examination of this area is needed.

Járdi et al. compared the coenology of the acidic sandy grasslands, steppes, meadows, and swamp meadows, which were grazed by Charolais and Hungarian gray cattle. The species pool and the cover of common species differed greatly in the examined grasslands, which clearly showed effects of the different abiotic and pedologic factors, water supply, and landscape uses. In these sandy grasslands, *Festuca ovina* L. aggregate (Poaceae) and *Festuca rupicola* Heuff. both appeared, and both *Festuca pseudovina* Hack. ex Wiesb. and *Stipa borysthenica* Klokov ex Prokudin were common in the more arid areas [25].

Examinations can be expanded with data from satellite images [26]. Sentinel-2A is the most useful satellite for this goal [27]. It was launched on June 23, 2015 as part of the European Copernicus Program.

Our questions were the following:

(i)     Do sand steppes and forest-steppes occur in the Ipoly Valley and if yes, then where?
(ii)    What kind of environmental effects influence the species composition on these areas?
(iii)   Besides classic habitat mapping, are the satellite data from Sentinel-2A useful for distinction of different areas?

## 2. Materials and Methods

The study area covers 445 hectares and is situated in northern Hungary along the River Ipoly, in the municipality area of Dejtár, and a smaller proportion between Dejtár and Ipolyvece (Figure 1).

Manual GPS was used for recording coordinates of the sample points with consideration to the edges of the habitat patches, which were treated as separate units. For the identification of habitats, the protocol of the Hungarian Habitat Classification System (ÁNÉR) was used. This system was developed in connection with the National Biodiversity Monitoring System (NBmR), which contains all vegetation types in Hungary [29]. It is the most frequently used complex system in the country and is under continuous development. Compared to coenological systems, NBmR is substantially simpler, as it contains fewer and broader categories. It groups associations into larger, more interpretable types and it is also feasible for practical use in nature conservation [30,31]. Association names were used according to [32]. Species nomenclature was used according to [33].

In general, one demarcated habitat patch belongs to only one habitat type (e.g., P2b see below in Table 1). However, there were habitat patches in which more than one habitat type was present. The main reason for this was that in some cases, habitat patches could not be allocated from the habitats in presentable size or they appeared as a mosaic of two or more ÁNÉR categories, therefore these habitat patches were indicated as habitat complexes in decreasing order of share in the following manner: D34 × B1a × P2a. The relevant ÁNÉR categories are shown in Table 1. While editing the habitat map, it was important to clearly trace and evaluate changes of habitat types.

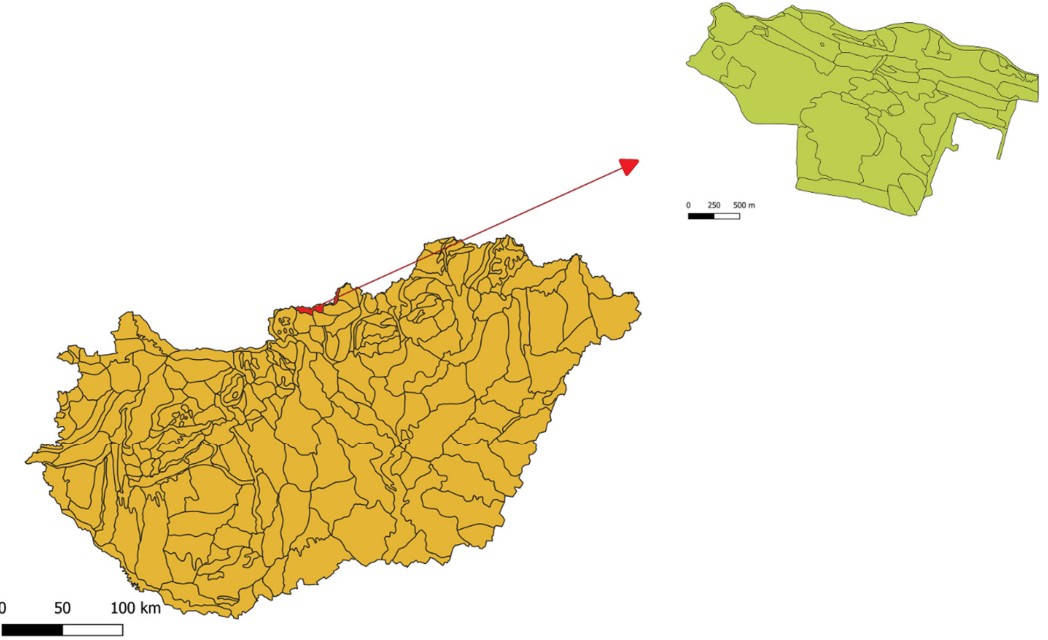

**Figure 1.** The location of the study area in Hungary (prepared with Marosi and Somogyi 1990) [28].

**Table 1.** Habitats revealed in the study area, using the categories of the Hungarian Habitat Classification System (ÁNÉR).

| ÁNÉR Categories | Description |
| --- | --- |
| B1a | Eu- and mesotrophic reed and Typha beds |
| B5 | Non-tussock tall-sedge beds |
| D34 | Mesotrophic wet meadows |
| H5b | Closed sand steppes |
| P2b | Dry and semi-dry pioneer scrub |
| P2a | Wet and mesic pioneer scrub |
| J3 | Riverine willow scrub |
| J4 | Riverine willow-poplar woodlands |
| OB | Uncharacteristic mesic grasslands |
| OC | Uncharacteristic dry and semi-dry grasslands |
| P2b | Dry and semi-dry pioneer scrub |
| RB | Uncharacteristic or pioneer softwood forests |
| S2 | Populus × euramericana plantations |
| S4 | Scots and black pine plantations |
| T1 | Annual intensive arable fields |
| U8 | Water streams |
| U7 | Sand, gravel, clay and peat mines, loess walls |
| U9 | Standing waters |
| U11 | Roads and railroads |

Maps were created using QuantumGIS, an open source geoinformatical program, which can be downloaded from (www.qgis.org). Coordinates recorded in the field were visualized as a GSX file in the program. The Satellite images chosen for evaluation were recorded on 17th September 2019 as the cloud cover was relatively low and this date was as close to the field research's date as possible. All Sentinel-2A image was downloaded from the official homepage of Copernicus. From the downloaded 12 optical bands, two optical bands were used to extract the Normalized Vegetation Index (NDVI) data. The visible red (RED) and near-infrared (NIR) were used to compute the numerical Normalized Vegetation Indices (NDVI) of each $10 \times 10$ m pixel [26], then the pixels were colored accordingly. NDVI is a non-dimensional value that reflects the vegetational activity of a given area. It is returned by the quotient of the sum and the difference of the reflected intensity of NIR and RED [34]. NDVI shows the biological activity of the vegetation: the higher the reflection of the chlorophyll, the higher the value. In the absence of vegetation, NDVI will be negative, for example on water bodies in the early vegetation period. In the evaluation phase habitats mapped by using the classic field survey method were compared with NDVI of satellite images.

In order to visualize the latter, 20 points were selected randomly to each ÁNÉR category, then the NDVI of the points were assigned to the categories. Data were analyzed using Microsoft Excel and PAST (PAleontological STatistics) software [35].

Mean NDVI data were compared by the non-parametric Kruskal-Wallis test, since raw data of 5 habitat types out of the 19 habitat types involved in this analysis did not fit the Gaussian distribution. Dunn's Multiple Comparisons test was used as a post hoc test, and differences at level $p < 0.05$ were considered significant [35].

## 3. Results

### 3.1. Habitat Map with ÁNÉR Categories

Based on the field works, 19 habitat types (i.e., ÁNÉR categories) and their combinations (habitat complexes) were identified, resulting in a total of 29 patch types in the map. Table 2 summarizes the size and the number of the identified habitat patches.

**Table 2.** The number and the size of the habitat patches identified in the study area.

| Type of Habitats | Number of Habitat Patches (Pieces) | Area (Hectares) |
|---|---|---|
| B1a | 4 | 46.0 |
| B2 × B5 | 1 | 17.5 |
| D34 | 1 | 34.1 |
| D34 × B1a × P2a | 1 | 16.5 |
| H5b | 16 | 38.0 |
| H5b × P2b | 2 | 28.1 |
| J3 | 1 | 6.8 |
| J3 × B5 | 1 | 1.0 |
| J3 × P2b × U9 | 1 | 0.9 |
| J4 | 2 | 0.5 |
| J4 × B5 | 1 | 3.3 |
| J4 × P2b | 7 | 25.7 |
| J4 × P2b × U9 | 1 | 2.7 |
| J4 × U8 | 2 | 11.3 |
| OB | 1 | 8.1 |
| OC | 8 | 30.6 |
| P2b | 15 | 18.0 |
| P2b × D34 | 1 | 1.1 |
| P2b × H5b | 1 | 13.1 |
| P2b × OC | 1 | 0.2 |
| RB | 2 | 84.9 |
| S2 | 5 | 30.4 |
| S4 | 2 | 3.5 |
| S7 | 2 | 6.2 |

**Table 2.** *Cont.*

| Type of Habitats | Number of Habitat Patches (Pieces) | Area (Hectares) |
|---|---|---|
| T1 | 2 | 7.1 |
| U11 | 3 | 6.8 |
| U6 | 1 | 0.4 |
| U7 | 1 | 0.7 |
| U9 | 2 | 1.7 |
| Total | 88 | 445.1 |

*3.2. Habitat Types of the Study Area*

According to the habitat map (Figure 2), mosaics of gallery forests (J4) and swamp meadows (B1a, B2, B5) are common in the north-eastern edge of the area. Wet meadows are also common due to the water supply. Figure 3 shows the vegetation belt in the foreground of the gallery forest. It is seemingly continuous and its parts are difficult to distinguish during habitat mapping. From the narrow belt of the gallery forest (Figure 3A1) the following plant associations appear: *Phragmitetum vulgaris* Soó 1927, *Scirpetum lacustris* Chouard 1924, *Typhetum angustifoliae* Pignatti 1953, *Typhetum latifoliae* Lang 1973, *Sparganietum erecti* Roll 1938. In the wetter part (Figure 3A2), swamp meadows and *Carex*-dominanted associations appear as a complex, with patches of different sizes: *Caricetum gracilis Almquist* 1929, *Caricetum vesicariae* Chouard 1924, *Galio palustris-Caricetum ripariae* Bal.-Tul.et al.1993, *Caricetum vulpine* Soó 1927, *Caricetum melanostachyae* Baláž 1943, *Caricetum distichae* Jonas 1933, and in some places even the *Phalaridetum arundinaceae* Libbert 1931, *Oenantho aquaticae-Rorippetum amphibiae* R.Tx.1953 and the *Butometum umbellati* Philippi 1973. In this belt, *Lychnis flos-cuculi* L. (Figure 3B) and *Ranunculus* ssp. are also dominant in the spring aspect.

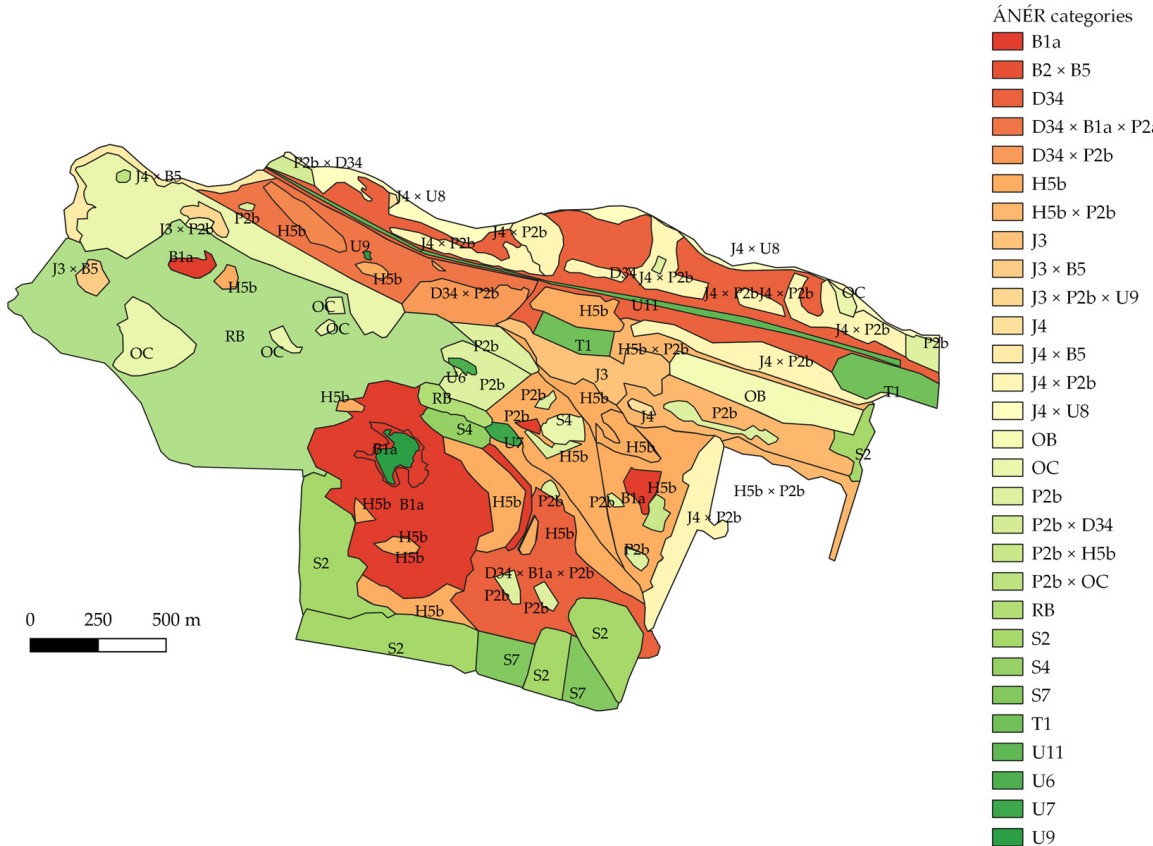

**Figure 2.** Habitat map of the Dejtár area using ÁNÉR categories.

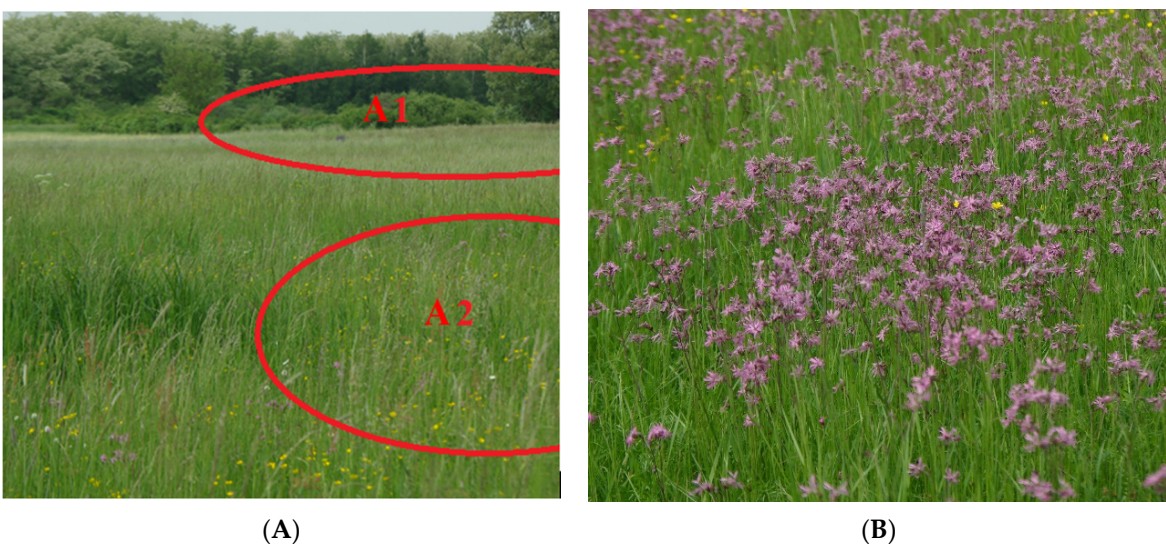

(**A**)                                                        (**B**)

**Figure 3.** Lower seated area of the Ipoly Valley. (**A1**): Gallery forest along the Ipoly; (**A2**): Swamp meadow dominated by *Carex* spp. (**B**): In the belt area in the Ipoly Valley, *Lychnis flos-cuculi* and *Ranunculus* ssp. are in the spring aspect.

On higher elevations, sandy grassland and shrubby, woody patches appear (Figure 4A). The number of patches of sand steppe meadows (H5b) was the highest (16 pieces) and its total area was also large (63.4 ha). However, considering the associations, they are less continuous than wet patches, because the drier parts of the higher elevations were dominated by *Stipa borysthenica*, which is an element of *Festucion vaginatae* Simon 2000.

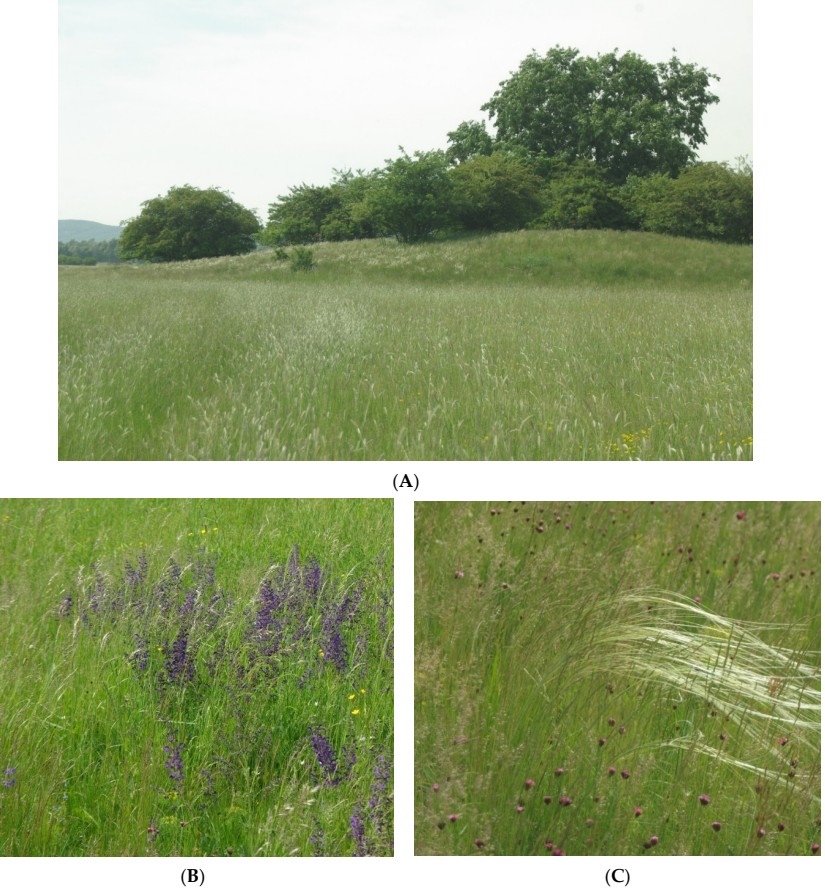

**Figure 4.** (**A**) Sandy grasslands and shrubs appearing in the higher seated parts. (**B**) *Salvia pratensis*, a dominant steppe species. (**C**) *Stipa borysthenica*, a dominant steppe meadow grass species.

In contrast, *Festuca vaginata* W. et K., an important characteristic species of open sandy grasslands is missing and the frequent occurrences of *Salvia pratensis* L. (Lamiaceae) are a sign of the steppe character and causes the vegetation to be akin to *Salvio-nemorosae-Festucetosum rupicolae* Zólyomi ex Soó 1964. From a coenological perspective, it is an interesting situation when a loess-steppe vegetation appears on sandy soil, with *Festuca rupicola* and *Salvia pratensis* as the dominant species (Figure 4B). *Thymus* spp., *Dianthus pontederae* A. Kern. and *Koeleria cristata* (Ledeb.) Schult., which are steppe elements, are also found here. Similarly to the original acidic Pannon sandy grasslands, *Pulsatilla pratensis* ssp. *nigricans* also appears. This vegetation type is very close to *Potentillo arenariae-Festucetum pseudovinae* Soó 1938, 1940, which is a rare association in the eastern part of the Pannonian region, as many of its species appear here.

Furthermore, in the most nutrient-poor parts of the area, diversity of the vegetation of sandy hedges is expanded by *Thymo serpylli-Festucetum pseudovinae* Borhidi 1958 as a new occurrence in the Pannon region. This association was known only from the eastern part of the Pannonian region (Nyírség), and was described in southwestern Hungary (Inner Somogy). Its important characteristic species is *Corynephorus canescens* (L.) P.Beauv. On sandy plains, these habitat patches appear along with pioneer arid and semiarid woody associations (P2b) as a mosaic. This vegetation forms after deforestation or as a consequence of heavy grazing.

*3.3. NDVI versus ÁNÉR Categories*

Lower vegetation productivity results in lower NDVI value [27]. In the study area (Figure 5), the lowest NDVI value was 0.076 while the highest was 0.83. Urban areas and water bodies have the lowest values, near 0. Still waters (ÁNÉR category: U9) have NDVI values between 0.08 and 0.55. The positive values can be explained by the biological activity in the water but the level of reflection is very low due the chlorophyll-poor areas [30]. Intensively cultivated farmlands (T1) have similar values. The category of roads and railroads (U11) showed low NDVI values similarly to the previous ÁNÉR categories due to the low biological activity; therefore, this category is well separated from the other classes. Swamp meadows (D34) on lower elevations along the river are more unified, while the reefs are distinctly different, which means that the differences originating from the elevation can also be observed in the vegetation of the area. NDVI values are higher where plant activity is higher, or the phenological phase of the plant is in the growing period. More arid sand associations showed low NDVI (0.56–0.632). Most sandy steppe meadows (H5b) along with transitional habitats between grasslands and woody patches had values of 0.632–0.666. Swamp meadow vegetation (D34) is clearly distinguishable from *Saliceto-Populetum* Meijer-Drees 1936 (J4) and dry-semi-dry pioneer shrubs (P2b) complexes along the Ipoly. Woody vegetation did not appear to be very unified, while NDVI categories differed depending on the phenological phase.

Figure 6 shows the different NDVI values of randomly chosen 20 pixels from each ÁNÉR category. Results of the statistical comparison of mean NDVI values among the habitat types are summarized in Table 3. Water bodies (U9) urban areas, such as fallow lands (T1) sand mines (U7), and roads (U11) differed significantly from the majority of other habitats. A somewhat higher but still low NDVI value was found in the sandy steppe meadows (H5b). Woody and grassland vegetation differed from each other. Categories of grassland vegetation, such as sandy steppe meadows (H5b), uncharacteristic arid and semiarid grassland complexes (OC), and uncharacteristic fresh grasslands (OB) showed lower NDVI values. In contrast, woody vegetation patches, such as riverine willow shrubs (J3), planted pinewoods (S4), and Riverine willow-poplar woodlands (J4) significantly differed from most of the non-woody habitats. Dry shrub vegetation with *Crataegus monogyna*, *Prunus spinosa* L. and *Juniperus communis* L. (P2b) showed uniformly medium high values. Wetter and arid grasslands, which are important in terms of grassland farming, differed greatly. Dry grasslands were not uniform because of the species characteristic in them, while in dry grasslands the cover of *Corinephorus canescens* (L.) P.Beauv., *Festuca*

*psedovina* Hack., *Festuca ovina*, *Festuca rupicola*, and *Stipa borysthenica* are larger. These species occur sporadically in the OC category.

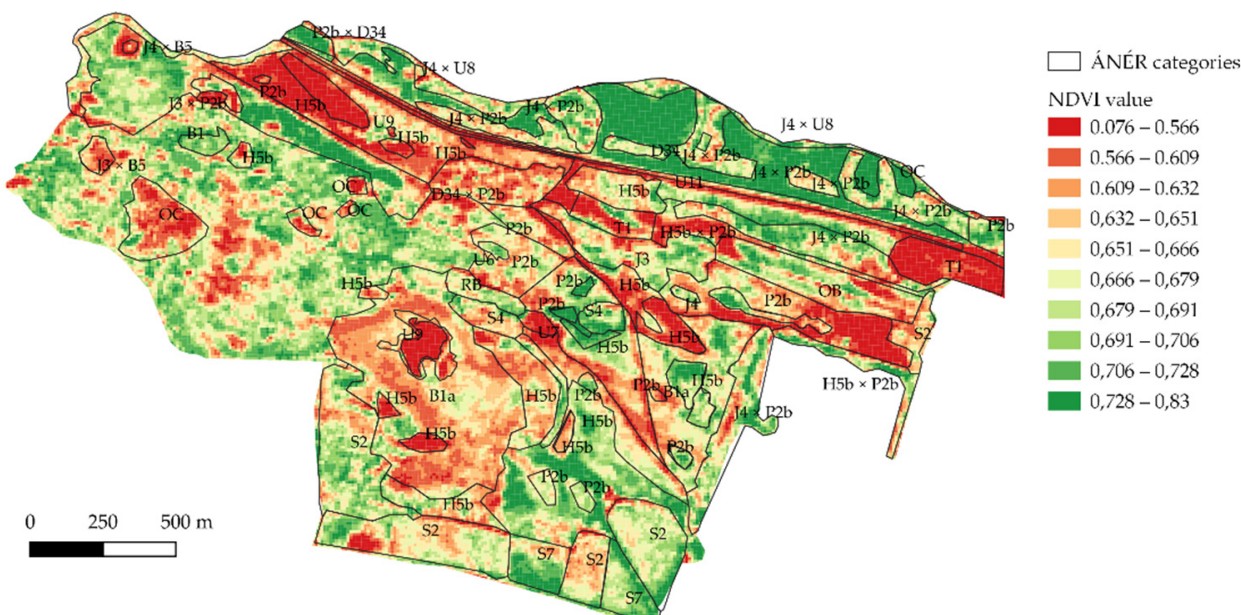

**Figure 5.** Habitat maps of Dejtár area using ÁNÉR categories and Normalized Vegetation Index (NDVI) data.

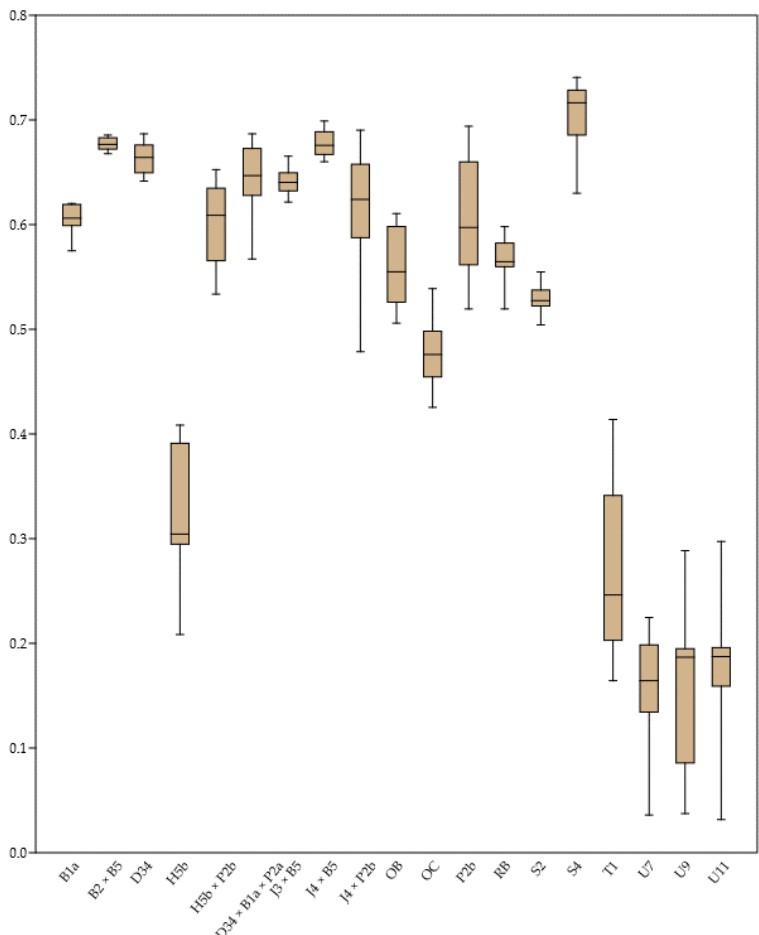

**Figure 6.** Distribution of ÁNÉR categories (habitat patches) based on NDVI data.

Table 3. Statistical comparison of mean NDVI values of the characteristic habitat types in the Ipoly Valley, Hungary. (*: $p < 0.05$, **: $p < 0.01$, ***: $p < 0.001$, ns: not significant).

| | B2 × B5 | D34 | H5b | H5b × P2b | D34 × B1a × P2a | J3 × B5 | J4 × B5 | J4 × P2b | OB | OC | P2b | RB | S2 | S4 | T1 | U7 | U9 | U11 |
|---|---|---|---|---|---|---|---|---|---|---|---|---|---|---|---|---|---|---|
| B1a | ns | ns | * | ns | ns | ns | ns | ns | ns | ns | ns | ns | ns | * | ** | *** | *** | *** |
| B2 × B5 | | ns | *** | ns | ns | ns | ns | ns | *** | *** | ns | ** | *** | ns | *** | *** | *** | *** |
| D34 | | | *** | ns | ns | ns | ns | ns | * | *** | ns | * | *** | ns | *** | *** | *** | *** |
| H5b | | | | * | *** | *** | *** | *** | ns | ns | ** | ns | ns | *** | ns | ns | ns | ns |
| H5b × P2b | | | | | ns | ns | ns | ns | ns | ns | ns | ns | ns | ns | ** | ** | *** | *** |
| D34 × B1a × P2a | | | | | | ns | ns | ns | ns | *** | ns | ns | * | ns | *** | *** | *** | *** |
| J3 × B5 | | | | | | | ns | ns | ns | ** | ns | ns | ns | ns | *** | *** | *** | *** |
| J4 × B5 | | | | | | | | ns | *** | *** | ns | ** | *** | ns | *** | *** | *** | *** |
| J4 × P2b | | | | | | | | | ns | * | ns | ns | ns | ns | *** | *** | *** | *** |
| OB | | | | | | | | | | ns | ns | ns | ns | *** | ns | * | ** | * |
| OC | | | | | | | | | | | ns | ns | ns | *** | ns | ns | ns | ns |
| P2b | | | | | | | | | | | | ns | ns | ns | *** | *** | *** | *** |
| RB | | | | | | | | | | | | | ns | *** | ns | ** | ** | ** |
| S2 | | | | | | | | | | | | | | *** | ns | ns | ns | ns |
| S4 | | | | | | | | | | | | | | | *** | *** | *** | *** |
| T1 | | | | | | | | | | | | | | | | ns | ns | ns |
| U7 | | | | | | | | | | | | | | | | | ns | ns |
| U9 | | | | | | | | | | | | | | | | | | ns |

## 4. Discussion

In the present vegetation study, we also discovered two vegetation types: the *Potentillo arenariae-Festucetum pseudovinae* and the *Thymo serpylli-Festucetum pseudovinae.* The two vegetation types were discovered in the Pannonian region as a new occurrence, which may have appeared as a result of environmental conditions [8,31]. The sand hills protruding from the river floodplain offered dry and nutrient-poor habitats where sand vegetation appeared and was able to form.

Vegetation edges were studied primarily in the central Great Plain of the Carpathian Basin [12,14]. Similar transitional conditions under similar environmental, water, and soil conditions may develop at the northern occurrence limits of these typical Pannonian vegetation types. In addition, two vegetation types were discovered in the Pannonian region as a new occurrence, which may have appeared as a result of territorial utilization, as in the eastern part of the basin (Nyírség) [31]. During the examination of the sandy area, two new occurrences of acerbic sand were found in the Pannonian region, which were only known from the western (Inner Plain), southwestern (Inner Somogy), and eastern (Nyírség) areas of the Pannonian region [36,37].

Among the dominant species such as *Corinophorus canescens* (L.) P. B., *Jasione montana* L., *Veronica* ssp.; however, the dominant species *Festuca vaginata* did not occur. The dominant taxon in the more closed stands was *Festuca pseudovina*, which also indicates the degradation of the vegetation [37–39]. The occurrence of *Festuca rupicola* suggests cooler environmental conditions and climatic effects [40].

Complex patches appear in the deeper areas, mostly characteristic of individual Hungarian habitats [22,41,42], and wet, swampy, marshy, or water-bound vegetation patches appear in the depressions on the site, which [43–45] may generally be typical but the position of the ground water table is especially important in the central area of the Carpathian Basin, which leads to the appearance of diverse and species rich vegetation [15,16,45]. Spots from the ÁNÉR-based habitat mapping showed agreement with the remote sensing data, which were also used as controls. Typical sandy grassland species such as *Stipa borysthenica* also occur at the heights of the northern sand steppes. In addition, isolated parts within each patch can be detected, which provide additional information about land use, which is also of practical importance for grazing, where land use is important in its maintenance, as well as in the mosaic-like and diverse vegetation.

Data from the Sentinel-2A satellite offer an opportunity for mapping natural habitats [46–48]. Based on our observations, it can be seen that the individual vegetation patches can be distinguished well based on the ÁNÉR categories, thus the field mapping is facilitated by the data of satellite images in hard-to-reach areas, as others noticed during their work [27]. When constructing an association-accurate map, the individual vegetation types are not clearly separated if there is no need for a habitat map of such accuracy and a simpler category system can be used to solve this problem, which [27] also received attention during the study. In contrast, the regularities are clearly outlined in the studied habitats and the agricultural area shows a homogeneous picture since after the harvest the open farmland (Category T1 in Figure 5) shows a lower NDVI value [49,50]. One can see that the wet and dry spots are markedly different in between the connected areas. The sandy grasslands are different associations on the bedrock, giving a mosaic-like picture (Category H5b in Figure 5) [46,51]. Isolated areas within each patch can also be identified as an area used by cattle as resting areas to provide additional information on land use that is also of practical importance for grazing. In order to preserve the original vegetation of the area [52], we need to protect the endangered and protected plants and also the vegetation types from river regulation, ploughing, and deforestation of the areas.

## 5. Conclusions

Answering question (i), the northern boundary of the Pannonian steppe-forest-steppe vegetation type is found in the study area, which is the valley of the Ipoly River. The vegetation type appears in the drier, acerbic sandy areas along the rivers, as well as new

occurrence patches, and a special coenosystemic vegetation mixture is formed. (ii) In the study area, the nearby sand ridges provided an environmental background to learn about the boundaries of the Pannonian sand steppe and forest-steppe. The dominant species of the steppe patches was *Festuca rupicola*, which plays a similar coenosystemic role in the formation of the vegetation to *Festuca vaginata* in the central parts of the Carpathian Basin. Based on the data, the appearance of *Festuca vaginata* can also be expected due to global climatic changes [53–55]. (iii) Based on the Sentinel-2A data, we have seen that urban areas are separated from natural habitats but the individual vegetation types are not clearly separated.

**Author Contributions:** Conceptualization, I.J., F.S., and K.P.; Data curation, E.S.-F. and N.P.; Formal analysis, Z.Z.; Methodology, I.J., D.F., and P.C.; Software, D.S.; Supervision, K.P.; Writing—original draft, I.J.; Writing—review & editing, G.P. and K.P. All authors have read and agreed to the published version of the manuscript.

**Funding:** The work was funded by OTKA K-125423 and NKFIH-1159-6/2019.

**Institutional Review Board Statement:** Not applicable.

**Informed Consent Statement:** Not applicable.

**Data Availability Statement:** The data presented in this study are available on request from the corresponding author.

**Acknowledgments:** Our studies received financial support from the Higher Education Institutional Excellence Program (NKFIH-1159-6/2019) awarded by the Ministry for Innovation and Technology and by OTKA K-125423.

**Conflicts of Interest:** The authors declare no conflict of interest.

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
