# Peer review of "Habitat Mosaics of Sand Steppes and Forest-Steppes in the Ipoly Valley in Hungary"

_forests, doi:10.3390/f12020135_

Round 1

Reviewer 1 Report

The topic of work is very important economically. Habitat mosaics of sand steppes and forest-steppes in the Ipoly Valley, Hangary were presented in the reviewed scientific work on 11 pages. The purpose of the work is clearly stated in the manuscript. The presented research material is sufficient, properly compiled, but there is no statistical analysis of the results. The applied research methods are adequate and properly described. The peer-reviewed scientific work has great utilitarian value and can be widely used in agricultural practice. Conclusions are missing from the reviewed scientific work.

- Figure 6 shows the Distribution of ÁNÉR categories based on NDVI data. The statistics of the results are missing, the authors write about the differences but do not show them.

- Conclusions and Author Contributions are missing from the work.

- Item 1 is not numbered in References. Moreover, items 4 and 5 are misspelled, they are also archaic. In References, in some places doi to or a link to the website are missing.

Author Response

Thank you for the comments and suggestions. Please see the attachment.

Reviewer 2 Report

I can't review this due to the very poor translation to English. There are just too many errors to even begin giving helpful feedback. Authors need to work with a professional translator to help them with this. 

Author Response

Thank you for your comments and suggestions. Please see the attachment.

Reviewer 3 Report

This manuscript deals with remarkable topics in the forest research, and although it is generally well-written, it is necessary to modify or add based on the contents of the 'Introduction' and 'Discussion' sections. The Introduction has the impression that the various information to elicit the hypothesis does not sufficiently reflect previous studies. In addition, In Discussion section, each argument is not explained based on empirical studies, so it feels abstract. So, I suggest that additional explanations are needed to support each topic or argument in the "Introduction and "Discussion" sections. The authors need to interpret and respond to the following comments in depth.

The Abstract should be reorganized according to the journal guidelines. The subtitle such as 'Background and Objectives' and 'Results' is not common in journal guidelines. Rather, this distinction gives the impression that it is disturbing to understand the totally outline of research. In addition, the numbers listed in the keyword must also be re-checked.

References inserted in "Introduction" and "Discussion" section should be re-checked in accordance with journal guidelines. When referring to a particular author, the author name and reference number shall be written together, and the location of the insertion of reference documents is the end of the relevant sentence.

Line 35. It requires a more detailed description of 'natural forest-steppe and steppe gain'. What plant species and communities have emerged and what characteristics are different or similar to other forests.

Line 43-45. Is the 'mosaic-like vegetation types' influenced by environmental variables? We are confident that we can explain this in more detail based on empirical studies. This may not just be a change in water level.

Line 48. How did previous research on 'Ipoly Valley' affect the distribution of plants to changes in water levels? We do not know what this means just by "influencing" in this paragraph. In general, changes of water level strongly affect the spatial distribution of aquatic plants types, such as emergent, free-floating, and submerged plants. We recommend explaining the previous investigation into 'Ipoly Valley' in more detail.

I don't think there are enough foregoing explanations to elicit this hypothesis. You need the following additional work to establish this hypothesis; i) for what reason did you select a this study site and what type of habitat does it lead to? ii) What gradients does the environmental characteristics of this area have and what effects does this have on plant distribution? Parts of Line 58-63, shall be organized on the basis of the above.

Figure 1. A detailed description of the two maps is required. What area map is it, and what does the letter shown on the map above mean? Also, what is the green area?.

Line 188-189. The findings show that the vegetation patches vary depending on the distribution of aquatic areas, such as wetlands, but more closely, the different distribution patterns of aquatic plants vary depending on the hydrological characteristics (water depth, pH, and etc.) of the aquatic ecosystem. If you insist that the mosaic pattern of the vegetation patches is closely related to the submerged area, it is necessary to add to the consideration how the distribution of aquatic plants varies depending on the gradient of the hydrologic characteristics of the aquatic ecosystem.

Line 197. What is the effect of two new occurrences of acerbic sand on vegetation patches?.

Line 201-204. If the occurrence of Festuca ruciola (Italic?) is related to the cold climate, does it mean that the area is getting cooler?. This result should be presented as a conclusion in this paragraph.

We think that the final conclusions of this study and its utilization methods are somewhat insufficient. How can changes in habitat patches affect this area? Empirical studies suggest that different distributions of plant communities determine efficiency as habitats for wildlife. In this sense, it is recommended that this results concludes in terms of biodiversity and management. Moreover, not only water levels and pH, but also other factors can affect the habitat patches. This can be further explained using existing references. Various explanations and interpretations of the results can induce the reader to use them in various ways.

Reference. Re-check your 'Reference List'. Refine the title, journal, volume, issue, etc. by referring to the journal form. The journal name is also abbreviated. Put a period on the back of the abbreviation. The number of Reference 1 is not listed. The reference number must match the text. Please take a closer look at this part.

Author Response

(The authors gave the same response as above.)

Reviewer 4 Report

The article presents research relevant from a local point of view. It presents a mapping of the Ipoly river valley, where there are sand steppe communities and a steppe forest. The first part lists the registered communities within the territory. The second part presents satellite data and the NDVI index. In both parts, the authors describe the current state at the time. The results represent a statement of the state and the fulfillment of the goals is not clear. E.g. how the authors of the one-off determination of NDVI want to identify the community when they themselves state that this affects a number of factors. The article, in my view, is too general, stating, without examining the dependencies between the elements. However, a lot of valuable data has been collected, so I recommend reworking the article and then publishing it. Other comments are given in the text in the attached file.

Author Response

(The authors gave the same response as above.)

Round 2

Reviewer 2 Report

Firstly, I think this paper has a lot of merit but is lacking focus and details. 

Not being familiar with the A-NER habitat classification, I am not aware of the current status of their GIS data layers. I was confused about whether you downloaded and used their maps in your study, or if you collected points in the field to generate your own maps. If you collected your own data, for readers unfamiliar with this process, you need to better explain how you did it. 

You need to provide more information about the satellite data - especially the resolution, and more information about NDVI - what is considered low, medium, high? You have a good idea about how these data could help delineate habitat polygons for inaccessible areas, but you need to better explain this process. 

You never really answer the questions about where are the borders of spread - you could done this with a map showing areas of transition - or the question about the usefulness of the satellite data.  

Author Response

Thank you for your suggestion and comments. Please see our answers in the attachment.

Reviewer 3 Report

I think this manuscript has been appropriately modified.

Author Response

Thank you for your approval!

Reviewer 4 Report

The authors largely respected the technical comments. What I appreciate. In terms of content, the paper is a mapping of the situation in a defined location. This certainly required an enormous effort. However, I am not convinced of the sufficient information and the subsequent possibility of using NDVI for the proposed purposes. The authors did not answer this question in the conclusions either. Maybe if only the descriptive characteristics of the area are left in the goals. It also has great value on its own.

Author Response

(The authors gave the same response as above.)
